# Insights into the Redox and Structural Properties of CoOx and MnOx: Fundamental Factors Affecting the Catalytic Performance in the Oxidation Process of VOCs

Veronica Bratan, Anca Vasile *, Paul Chesler * and Cristian Hornoiu

"Ilie Murgulescu" Institute of Physical-Chemistry of the Romanian Academy, 202 Spl Independentei, 060021 Bucharest, Romania

*   Correspondence: avasile@icf.ro (A.V.); pchesler@icf.ro (P.C.); Tel.: +40-21-318-85-95

**Abstract:** Volatile organic compound (VOC) abatement has become imperative nowadays due to their harmful effect on human health and on the environment. Catalytic oxidation has appeared as an innovative and promising approach, as the pollutants can be totally oxidized at moderate operating temperatures under 500 °C. The most active single oxides in the total oxidation of hydrocarbons have been shown to be manganese and cobalt oxides. The main factors affecting the catalytic performances of several metal-oxide catalysts, including $CoO_x$ and $MnO_x$, in relation to the total oxidation of hydrocarbons have been reviewed. The influence of these factors is directly related to the Mars–van Krevelen mechanism, which is known to be applied in the case of the oxidation of VOCs in general and hydrocarbons in particular, using transitional metal oxides as catalysts. The catalytic behaviors of the studied oxides could be closely related to their redox properties, their nonstoichiometric, defective structure, and their lattice oxygen mobility. The control of the structural and textural properties of the studied metal oxides, such as specific surface area and specific morphology, plays an important role in catalytic applications. A fundamental challenge in the development of efficient and low-cost catalysts is to choose the criteria for selecting them. Therefore, this research could be useful for tailoring advanced and high-performance catalysts for the total oxidation of VOCs.

**Keywords:** $CoO_x$; $MnO_x$; hydrocarbon total oxidation; redox properties; catalytic performances; lattice oxygen mobility





## 1. Introduction

### 1.1. General Features

According to the European Directive no. 1999/13/EC, a volatile organic compound (VOC) is defined as "any organic compound having at 293.15 K a vapor pressure of 0.01 kPa or more, or having a corresponding volatility under the particular conditions of use". Due to their high volatility, VOCs are considered the main pollutants in the air. They are emitted from oil and gas fields, in diesel exhaust, and they are also encountered in many home activities, such as painting, cooking, grass cutting, etc. These compounds are generally toxic and can cause eye irritation, respiratory problems, and even cancer [1–4].

In addition to the direct harmful effects of these compounds on human health, VOCs also have an indirect effect: they contribute to atmospheric pollution, being the precursors of ground-level ozone and photochemical smog, in the presence of $NO_x$ and solar radiation [1,5–7].

Photochemical smog is a mixture of nitrogen oxides and volatile organic compounds which are able to react under the action of sunlight:

$$NO_2 + UV\ radiation = NO + O \tag{1}$$

The resulting oxygen radicals further react with oxygen in the air to form ozone.

$$O + O_2 = O_3 \tag{2}$$

Atmospheric ozone can be dangerous for people (it is irritating and causes respiratory problems, including asthma) and can also attack certain materials, such as textiles, paints, works of art, books, etc. However, if $NO_x$ particles were the only pollutants in the atmosphere, ozone, resulting as shown in Equation (2), would be consumed in the reaction with NO, according to reaction (1):

$$O_3 + NO = NO_2 + O_2 \tag{3}$$

In this case, the ozone concentration in air will reach an equilibrium value and no longer be dangerous. However, if hydrocarbons (HCs) are also present in the atmosphere, part of the ozone reacts with them and produces toxic products, such as peroxyacetyl nitrate (PAN). PAN, in addition to having harmful effects on human health, reduces and may even stop the growth of plants, and in polluted regions can act as a source of $NO_x$ [8].

Depending on the nature of the functional group, the main volatile organic compounds emitted are classified into: aliphatic hydrocarbons (alkanes, alkenes, and alkynes), aromatic hydrocarbons, oxygenated organic compounds (alcohols, ketones, and esters), and halogenated organic compounds. Their impact on the environment and, implicitly, on human health, depends both on the nature of the VOCs and on their concentration in air. Unsaturated and aromatic hydrocarbons are the most polluting, especially due to their large contribution to the formation of "photochemical ozone". Thus, according to reference [5], in 2010, in China, alkenes/alkynes and aromatic hydrocarbons accounted for 28% and 54%, respectively, of the total VOCs with ozone-forming potential, although their effective contributions to the total emissions of organic compounds was only 8.5% and 34%, respectively.

*1.2. VOC Removal: General Methods and Materials*

Due to their harmful effects on human health and the environment, VOC abatement has become imperative. Nowadays, there are several methods generally used to eliminate VOCs from the air stream, such as adsorption onto activated carbons or zeolites, thermal combustion, and photocatalytic and catalytic oxidation [6,9–13]. The emission sources for volatile organic compounds, as well as the variety of compounds emitted by each source, are so diverse that all disposal methods have practical limitations. Adsorption is restricted to highly diluted VOC emissions and is limited by the difficult recovery of the adsorbent [12,14]. Thermal combustion needs high temperatures in order to achieve full oxidation of concentrated VOC streams (higher than 800 °C), making it not very economical. Furthermore, incomplete thermal oxidation of VOCs can produce numerous harmful byproducts (dioxins, nitrogen oxides, etc.) [2,14]. Photocatalytic oxidation works at low temperatures, but it has lower efficiency and can also produce harmful byproducts [1].

Thus, catalytic oxidation remains a better alternative, as the pollutants can be totally oxidized at lower operating temperatures (200–500 °C). Recently, high-performance catalysts have been developed for the removal of pollutants at low temperatures. In general, by the deposition of noble metals on various oxide supports, superior performances in VOC catalytic oxidation have been achieved [15–20]. Some problems with using noble-metal-based catalysts are their high cost and their poor resistance to poisoning, hence the need to replace these materials with abundant ones which are cheaper, less susceptible to supply fluctuations, and more environmentally friendly.

Transition metal oxides are industrially important materials, and many experimental studies have been conducted on the oxidation of VOCs over them. The most commonly used metal-oxide catalysts include Cr, Mn, Co, Ni, Fe, Cu, and V, used pristine or mixed, with or without supports [1,13,14,20–27]. Their behaviors in oxidation reactions of hydrocarbons have been studied for many years. Blazowski et al. [28] found that $Co_3O_4$, $Cr_2O_3$,

CuO, NiO, MnO$_2$, and V$_2$O$_5$ presented good performances in catalyzing the oxidation of hydrocarbons. Dixon et al. [29] stated that, among the metal oxides corresponding to the fourth-period transition-metal series, higher activity was presented by Cr$_2$O$_3$ and Co$_3$O$_4$. Moro-oka et al. [30] also reported that the highest rate of reaction in the oxidation of several hydrocarbons (acetylene, ethylene, propane, and isobutene) over a series of metal oxides (Co$_3$O$_4$, NiO, MnO$_2$, Fe$_2$O$_3$, Cr$_2$O$_3$, and CeO$_2$) was found for Co$_3$O$_4$ and MnO$_2$. However, the best catalytic performances in the latter study were obtained for Pt and Pd catalysts.

Although they are generally less active than noble-metal catalysts, they have the advantages of much lower costs, higher thermal stabilities, and resistance to poisoning, which are ultimately reflected in a decrease in the total cost of the depollution technology. Recent studies have confirmed that the most active single oxides in the total oxidation of hydrocarbons are manganese and cobalt oxides [31–38]. Hence, Lahousse et al. showed that the $\gamma$-MnO$_2$ catalyst outperformed the 0.3 wt% Pt/TiO$_2$ catalyst in the oxidation of n-hexane [39], and Lin et al. [40] synthesized a mesoporous $\alpha$-MnO$_2$ microsphere with high toluene-combustion activity comparable to that of a Pd-based FeCrAl catalyst. Among Co oxides, the most active catalyst in the complete oxidation of hydrocarbons has been shown to be Co$_3$O$_4$. The studies performed in references [41,42] showed better performances in the propane oxidation reaction of Co$_3$O$_4$, even when it was compared with some commercial as well as self-made Pt, Pd deposited on different support catalysts (Figure 1).

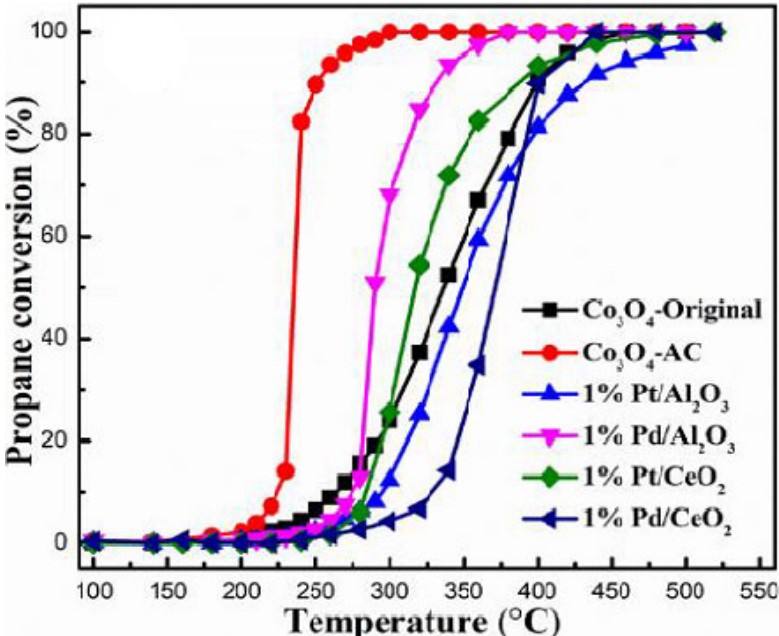

**Figure 1.** Catalytic combustion of propane (0.3% C$_3$H$_8$, 10% O$_2$, and N$_2$ balanced) as a function of temperature over Co$_3$O$_4$, commercial 1%Pt/Al$_2$O$_3$ and 1%Pd/Al$_2$O$_3$, and self-made 1%Pt/CeO$_2$ and 1%Pd/CeO$_2$, at WHSV of 240,000 mL g$^{-1}$ h$^{-1}$. Reprinted and adapted with permission from Ref. [41]. Copyright 2018 Elsevier.

In this review, we mainly focus on the complete catalytic oxidation of hydrocarbons at moderate temperatures (below 500 °C) using transition metal oxides, namely, cobalt and manganese oxides. The review will be divided into three parts. First, a brief description of the reaction mechanisms is presented. The second and third parts deal with CoO$_x$ and MnO$_x$ oxides, respectively. In both cases, we examined these single-metal oxides that are widely studied as active components in the catalytic oxidation of VOCs and investigated some important influencing factors affecting their catalytic performances. As an important characteristic of the metal-oxide catalysts, the mobility of oxygen species and redox properties has been emphasized. Several case studies have been presented to illustrate the correlations between structure and morphology, textural properties (surface

area and porosity), the nature of the exposed facets and crystal defects, redox properties, and catalytic activity. Finally, we present the conclusions and our perspectives on future developments in this field.

## 2. Reaction Mechanism Overview

A fundamental challenge in the development of efficient and low-cost catalysts for the removal of pollutants from the air is to first understand and then choose the main criteria for selecting them. This can be undertaken as a strategy for tailoring new catalysts with improved structures and properties. For this purpose, the most important step is to understand the reaction mechanism. Three kinetic models are usually used to describe the mechanism for the complete oxidation of hydrocarbons: the Langmuir–Hinshelwood (L–H), the Eley–Rideal (E–R), and the Mars–van Krevelen (MvK) models.

Langmuir was the first to describe how a bimolecular reaction takes place at the surface of a catalyst [43,44]. He identified two types of surface interactions: (i) interaction between molecules or atoms adsorbed at adjacent sites on the surface, which is known as the Langmuir–Hinshelwood (L–H) mechanism; and (ii) an interaction that takes place as a result of the collision of the gas molecules of one of the reactants and the adsorbed molecules of the other, which is known as the Eley–Rideal (E–R) mechanism (Figure 2).

Therefore, in the L–H model, both the VOC and oxygen molecules are adsorbed on the surface of the catalyst. The two reactant molecules could be adsorbed at different sites or they could compete for the same site. After the reaction between adsorbed reactants takes place, the products ($CO_2$ and $H_2O$) desorb from the surface of the catalyst. In the Eley–Rideal (ER) mechanism, oxygen is too weakly adsorbed, so that it reacts directly from the gas phase with adsorbed VOC, or the inverse [45].

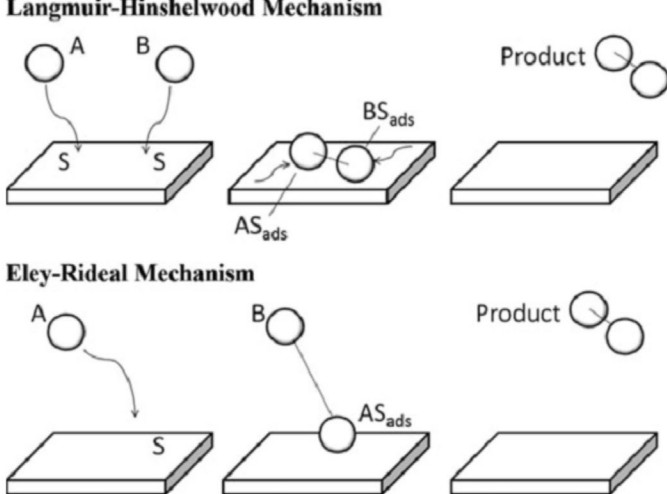

**Figure 2.** Top model: The Langmuir–Hinshelwood mechanism: two molecules adsorb onto the surface and diffuse and interact with each other until a product is formed and desorbs from the surface. Bottom model: The Eley–Rideal mechanism: a molecule adsorbs onto the surface and another molecule interacts with the adsorbed one until a product is formed and desorbs from the surface. Reprinted with permission from Ref. [46]. Copyright 2014 Springer Nature.

Almost at the same time as the formulation of these two mechanisms, another one emerged based on the idea that the lattice components of the catalyst appear in the reaction products [47–49]. In 1954, Mars and van Krevelen [50] proposed this new mechanism, and now it is universally accepted as being the one that describes the oxidation of hydrocarbons on metal-oxide catalysts at moderate temperatures [3,14,51]. According to this mechanism, denominated the Mars–van Krevelen or redox mechanism, the reaction involves two consecutive steps.

In the first step, the hydrocarbon adsorbed on the surface of the catalyst is oxidized by the oxygen atom from the oxide network. As a consequence, an oxygen vacancy (–□–) appears on the surface, leaving the catalyst's surface in a reduced state.

$$C_xH_{y(ads)} + \text{- } M^{n+} \text{ - } O^{2-} \text{ - } M^{n+} \text{ - } \rightarrow xCO_{2(g)} + y/2H_2O_{(g)} + \text{- } M^{(n-1)+} \text{ - } \square \text{ - } M^{(n-1)+}, \quad (4)$$

In the second step, re-oxidation of the catalyst's surface takes place. Re-oxidation actually means the filling of the oxygen vacancy generated according to Equation (4). This could happen either directly, through the oxygen in the gas phase [52], following the equation:

$$\text{-}M^{(n-1)+}\text{-}\square\text{ -}M^{(n-1)+}\text{- } + 1/2O_2 \rightarrow \text{-}M^{n+}\text{-}O^{2-}\text{---}M^{n+}, \quad (5)$$

or indirectly, through the diffusion of the oxygen atom from the bulk of the oxide to the reduced site [53]. Thus, the mobility of the oxygen in the catalyst has to be high enough that an oxygen ion from the lattice can diffuse toward the newly formed anionic vacancy.

In view of these facts, the catalytic performance of a metal oxide in the total oxidation reaction of hydrocarbons will be dictated by:

(1)   Its ability to adsorb hydrocarbons;
(2)   Its redox properties (the ease of the reduction and reoxidation of its surface);
(3)   The mobility of the oxygen atoms in the oxide lattice.

The adsorption of a HC on the catalyst's surface depends both on the catalyst's properties and the HC's characteristics. For a metal oxide to be easily reducible, it has to possess the capacity to easily release an oxygen atom from the lattice. By releasing an oxygen atom, the metallic ion changes towards a lower valence state, leaving the oxide stable; this is characteristic mainly of transitional-metal oxides with metallic ions in high oxidation states. In addition, the reducibility of oxides could be improved if oxygen vacancies are present. The presence of an oxygen vacancy makes the adjacent lattice oxygen more easily released and/or transferred.

On the other hand, a good catalyst must also be readily re-oxidized, otherwise the reaction will not advance. In this respect, the presence of oxygen-vacancy defects is also an important factor because they favor the activation of gaseous oxygen (which is usually adsorbed on the anionic vacancies near reduced metallic ions, $M^{(n-1)+}$), according to the following reaction [54]:

$$O_{2(g)} \; \rightleftarrows \; O_{2(ads)} \; \underset{-e}{\overset{+e}{\rightleftarrows}} \; O_{2(ads)}^{-} \; \underset{-e}{\overset{+e}{\rightleftarrows}} \; 2O_{(ads)}^{-} \; \underset{-2e}{\overset{+2e}{\rightleftarrows}} \; 2O_{(ads)}^{2-} \; \rightleftarrows \; 2O_{(lattice)}^{2-}, \quad (6)$$

The ionic forms of oxygen which are stable on the metal-oxide surface, $O_2^{-}$ and $O^{-}$, are strong electrophilic species and could be themselves directly involved in the oxidation reaction of hydrocarbons by causing the breakage of C-C bonds [55]. In the temperature range of 100–300 °C, complex surface oxidation/re-oxidation mechanisms are operative [56,57] because of the lower activation energies of processes that involve adsorbed oxygen instead of lattice oxygen. The predominant mechanism will be established by the oxygen-storage capacity of the oxide and the oxygen-lattice availability.

## 3. Cobalt Oxides: The Influence of Structural and Redox Properties on Catalytic Performance

$Co_3O_4$ (with the extended formula $Co^{2+} Co^{3+}{}_2O_4$) has a spinel-type crystal structure in which $Co^{2+}$ occupies 8 tetragonal sites, $Co^{3+}$ takes 16 octahedral sites, and 32 sites are occupied by O ions (Figure 3). The lattice oxygen is cubically close-packed in a unit cell, with 1/8 of the tetrahedral sites occupied by $Co^{2+}$ and half of the octahedral sites occupied by $Co^{3+}$ [58].

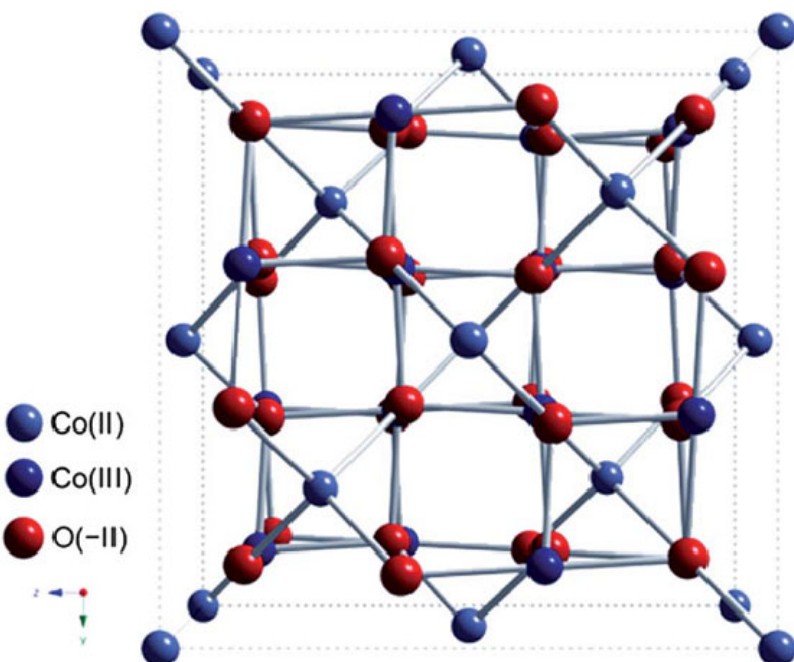

**Figure 3.** Scheme of the spinel structure of cobalt(II, III) oxide. Reprinted with permission from Ref. [58]. Copyright 2012 Royal Society of Chemistry.

Although cobalt oxides are the most efficient oxides used as catalysts in the total oxidation of pollutants in the atmosphere, at low temperatures a disadvantage in their use is their deactivation during the reactions. The deactivation occurs either at low temperatures (below 100 °C) under reaction conditions (since the reoxidation of the catalyst is assumed to take place slowly), or at temperatures above 500 °C, when sintering of the oxide particles occurs, producing a significant decrease in the specific surface area and, implicitly, a decrease in catalytic performance. Liu et al. [59] observed an accelerated deactivation after cycling of the reactor temperature between 210 and 500 °C. However, it is worth noting that in the second cycle no further deactivation occurred. Therefore, by working at intermediate temperatures, these shortcomings could be prevented.

Zhang et al. [60] prepared $Co_3O_4$ catalysts via a simple precipitation method using various precipitants or precipitant precursors: oxalic acid, sodium carbonate, sodium hydroxide, ammonium hydroxide, and urea. A commercial $Co_3O_4$ catalyst (Co-com) was also investigated for comparison purposes. They studied the influence of various redox and structural properties on the performances of the obtained samples in the catalytic oxidation of propane and toluene. The sample synthesized using carbonate had the highest catalytic activity, and this was related to the enhancement of the surface area, the number of lattice defects, surface $Co^{2+}$ concentration, and the reducibility of the sample (Figure 4). In fact, although the catalyst Co-$CO_3$ exhibited a specific reaction rate for propane oxidation much higher than that of other catalysts, its performance was comparable to that of the Co-com catalyst, the latter having a larger specific surface area that overcompensated its lower intrinsic activity. However, Co-$CO_3$ proved an excellent ability to oxidize toluene and propane completely, while Co-com exhibited lower reducibility, which negatively affected the catalytic performance in toluene oxidation.

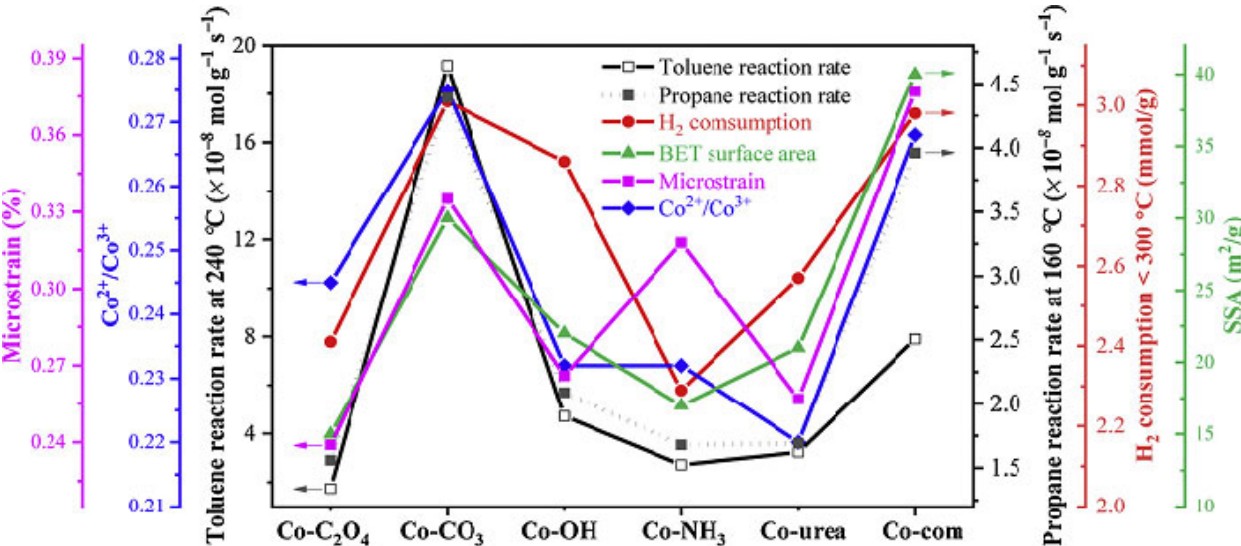

**Figure 4.** The relationship between the catalytic reaction rate and several physical and chemical parameters. Reprinted with permission from Ref. [60]. Copyright 2020 Elsevier.

The experimental studies focused on synthesizing cobalt oxides with small particle sizes, large specific surface areas, and different morphologies in order to obtain materials with large numbers of active sites, high concentrations of surface defects (especially oxygen vacancies), and better reducibility—factors which could improve these oxides' performances as oxidation catalysts.

Solsona et al. [61] successfully obtained $Co_3O_4$ nanoparticles and then tested them in propane total oxidation. The decrease in particle size resulted in an increase in specific surface area—a paramount factor in heterogeneous catalysis. Among the studied $Co_3O_4$ catalysts, those with the higher surface areas were the most reactive, total conversion being obtained at a reaction temperature lower than 250 °C. The authors found that the small size of the crystallite had a beneficial effect on catalytic performance: not only did it have a large number of active sites (the larger the exposed surface area, the greater the number of accessible active centers), it also produced a decrease in the energy of the Co-O bonds, which meant that oxygen could be easily released, increasing the oxide reducibility of cobalt oxide.

Other authors also found that the activity of Co catalysts in the total oxidation reaction of propane was mainly determined by the specific surface. Puertolas et al. [55], studying several $Co_3O_4$ oxides obtained by a hydrothermal method in the presence of various organic acids, observed a decrease in surface area compared to the reference sample (prepared without the addition of organic acids). Although the reasons for this decrease were not completely explained, it was found that the reference sample had better catalytic performances than the oxides obtained in the presence of organic acids. As the specific reaction rates (normalized on the surface) calculated for all of the studied cobalt-oxide catalysts had almost the same values, the authors assumed that the bulks and surfaces of the catalysts possessed similar redox characteristics. In conclusion, it was stated that the differences in catalytic performances were mainly determined by the different surface areas, establishing that there is a direct relationship between specific surface area and $Co_3O_4$ activity.

In this regard, mesoporous transition metal oxides could be prominent candidates for use as catalysts. They possess large specific surface areas but also ordered pore structures, which could facilitate the diffusion of reactant molecules towards active centers where they can be adsorbed, which is the first important step in the enhancement of catalytic performance. This is why numerous studies have been performed in order to obtain mesoporous materials with improved physicochemical properties [35,42,62,63]. Thus, it was

possible to obtain $Co_3O_4$ oxides with specific surfaces of over 100 $m^2$/g, sometimes even higher than 300 $m^2$/g [35,63], and 3D ordered mesoporous structures, usually generated using mesoporous silica (KIT-6, SBA-15, or SBA-16) as a hard template. The as-prepared $Co_3O_4$ catalysts exhibited exceptional catalytic properties, reaching a $T_{90}$ (the temperature at which the conversion reaches a value of 90%, used for measuring catalytic performances) lower than 200 °C for the total oxidation of toluene, for example [63].

Another important factor that influences catalytic activity is the oxidation state of metallic ions. In reference [64], the authors followed the effect of the oxidation states of metallic ions by studying the oxidation reaction of xylene over three mesoporous cobalt-oxide catalysts with various $Co^{3+}$/$Co^{2+}$ ratios. It was found that the conversion measured at 240 °C increased in the order: $Co_3O_4$-CoO mixture > $Co_3O_4$ > CoO. The authors stated that, to obtain a high conversion, the presence of ions in a high oxidation state is necessary, but that better performances also require surface $Co^{2+}$ species, which could be slightly oxidized to $Co^{3+}$ species by oxygen gas (Equation (5)). The first step in the activation of gaseous oxygen on the oxygen vacancies near $Co^{2+}$ is the chemisorption of oxygen (Equation (6)), resulting in electrophilic adsorbed oxygen ($O_2^-$)—a very active species involved in the oxidation of VOCs. Liu et al. [65] also found that the extra CoO phase is beneficial for catalytic activity, decreasing the crystalline size and increasing the specific surface area and the $Co^{2+}$/$Co^{3+}$ and $O_{ads}$/$O_{latt}$ ratios (where $O_{ads}$ is adsorbed oxygen and $O_{latt}$ refers to the oxygen atoms from the oxide lattice).

In reference [66], the authors investigated the effects of the coexistence of cobalt defects and oxygen vacancies on the performances and mechanisms of a $Co_{3-x}O_{4-y}$ catalyst for the toluene-degradation reaction. Their studies proved that cobalt defects are favorable for the formation of oxygen vacancies that can accelerate the conversion of intermediates, thus obtaining over 90% toluene conversion at a temperature lower than 190 °C (300 ppm VOC, GHSV = 72,000 mL $g^{-1}$ $h^{-1}$).

Traditionally, reducing the size of particles is the main goal, but in recent years oxide catalysts with desired structural and physicochemical parameters have been obtained by controlling their morphologies by synthesis. For example, obtaining $Co_3O_4$ in the form of halospheres is a method of increasing the specific surface area. Such structures have the advantages of a very high surface area–volume ratio, low density, and high permeability for reactants [58,67]. At the same time, a particular morphology allows the predominant exposure of a certain facet with higher reactivity, which has a significant impact on catalytic performance. Thus, different morphologies of $Co_3O_4$, such as octahedra [68], cubes [68–70], rods, sheets [69,70], plates [69], etc., have been obtained and studied in the total oxidation of hydrocarbons. For this purpose, the conditions of the synthesis reaction, the precursor, the heat treatment, the surfactant, etc., could be varied (Figure 5).

Generally, the experimental studies on the oxidation of hydrocarbons (except methane) over $Co_3O_4$ have shown that the facet {110} is more active than other facets [70–72]. This result was explained by the higher mobility of oxygen vacancies on these planes, which permit two vacancies easily meeting together to react with $O_2$ [73]. Meanwhile, the {110} facet consists mainly of $Co^{3+}$ cations [69,70].

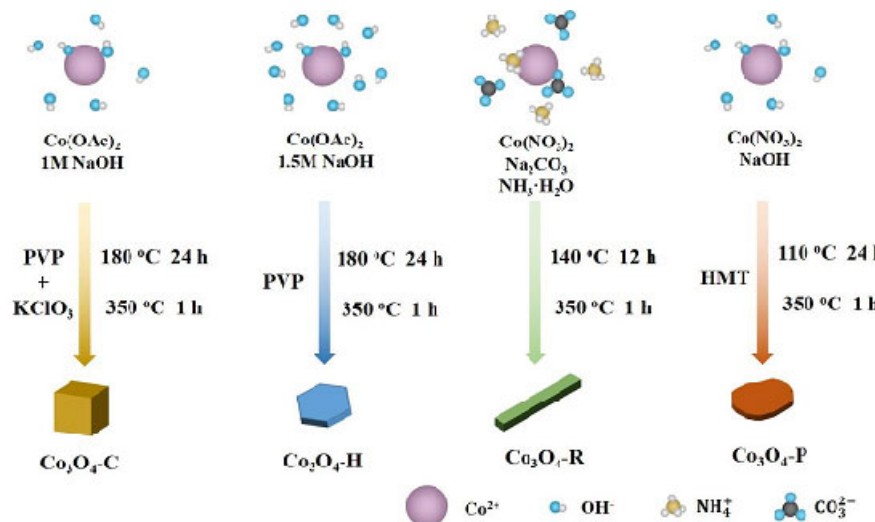

**Figure 5.** Schematic illustration of the synthetic routes for the investigated $Co_3O_4$ catalysts. Reprinted with permission from Ref. [69]. Copyright 2019 John Wiley and Sons.

Thus, in reference [70], $Co_3O_4$ rods, sheets, and cubes were synthesized with the exposed facets {110}, {111}, and {100}, respectively (Figure 6). Using various experimental techniques and theoretical calculations (by DFT), the authors comprehensively studied the behavior of the prepared catalysts in propane combustion, and they related the improved catalytic activity of the rod-type $Co_3O_4$ (Figure 6) to the higher reactivity of the predominantly exposed facets {110}. The facet {110}, mainly exposed on $Co_3O_4$ rods, had the highest calculated area and a higher density of low-coordination Co atoms, which can promote the generation of active oxygen. From DFT studies, the authors determined that this facet also had the lowest energy for oxygen-vacancy formation, which is beneficial for the adsorption and activation of $O_2$. This was experimentally confirmed by $O_2$-TPD and TPR, from which results it was confirmed that $Co_3O_4$-R had the most active electrophilic oxygen species ($O^-$) and was the more reducible sample.

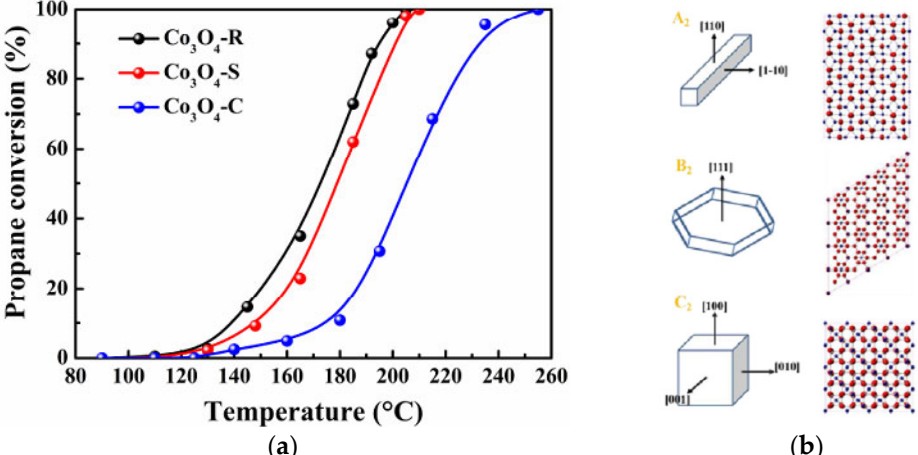

**Figure 6.** (**a**) Ignition curves of different catalysts for propane oxidation. (**b**) Corresponding geometric models of $Co_3O_4$ (A2, B2, and C2) and surface atomic structure models of (110), (111), and (100) facets. Reprinted and adapted with permission from Ref. [70]. Copyright 2021 Elsevier.

Electrophilic oxygen species, such as $O^-$ and/or $O_2^-$, are thought to be easily formed on the surface of $Co_3O_4$ and could be incorporated into the oxide lattice to replace the oxygen atoms which oxidize the hydrocarbons. Additionally, such reactive oxygen species are expected to directly act through an electrophilic attack on the C=C or C-H bonds, playing a decisive role in the catalytic degradation of hydrocarbons [74]. Several in situ studies have shown that propene/propane [75] and toluene [76] oxidations over $Co_3O_4$ follow a typical Mars–van Krevelen type mechanism. However, in the latter study, Zhong et al. [76] investigated the oxidation of toluene over $Co_3O_4$ through in situ DRIFTS combined with quasi-in situ XPS and UV–Vis diffuse reflectance spectroscopy. They proposed a redox mechanism in which toluene and oxygen molecules are firstly adsorbed on the oxide surface and oxygen is activated via oxygen vacancies. Afterward, the activated toluene molecules react with both the lattice and chemisorbed oxygen species, forming oxygenated intermediates which finally desorb as $CO_2$ and $H_2O$. Although the study revealed the decisive role of surface lattice oxygen, the authors also identified the importance of the presence of gaseous oxygen in the mineralization of the intermediate products and claimed surface adsorbed oxygen as active oxygen.

Liu et al. [77] also stated that the catalytic performances were related to the reactivity of the preferentially exposed crystallographic planes. Three $Co_3O_4$ samples were prepared using urea and three different cobalt precursors. The as-synthesized samples possessed different morphologies: hydrangea-like (H), spiky (S), and pompon-like (P) (Figure 7). The best catalyst, sample H, performed a 90% toluene conversion at about 248 °C, which was 60 °C lower than that of sample P (Figure 8). Its higher catalytic activity was mainly explained by the more active {110} exposed planes, in contrast to the P sample which had exposed {111} facets. Compared with the S sample, which possessed the same primarily exposed facets, the enhanced activity of H could be attributed to the larger number of active sites provided by its larger surface area (Table 1). Catalytic activity varies in line with the $O_{ads}/O_{latt}$ ratio (obtained via XPS) and with the number of oxygen vacancies (determined by Raman analysis). Meanwhile, the H sample contained a higher concentration of $Co^{3+}$.

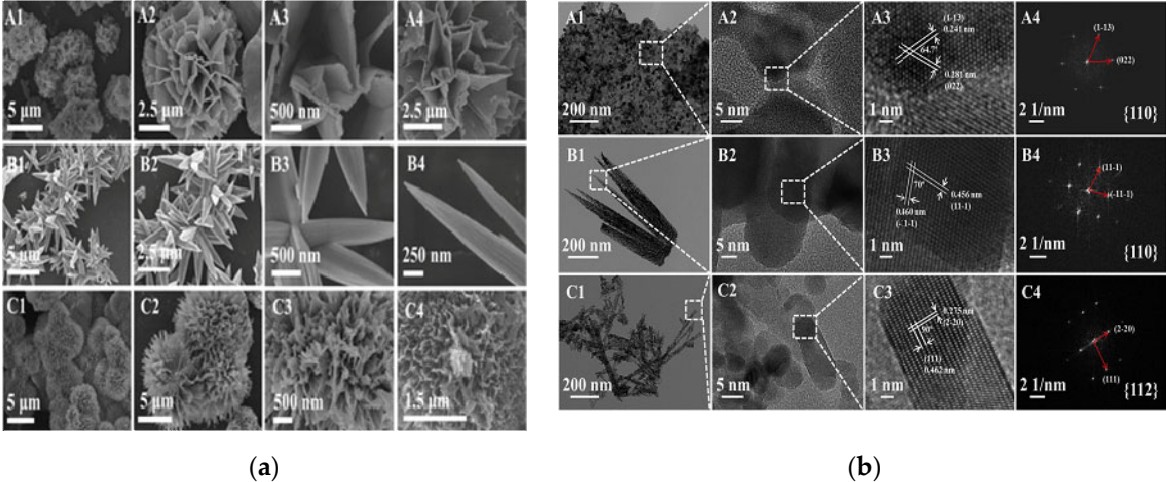

(a)                                       (b)

**Figure 7.** (**a**) SEM images of (A1–A3) H, (B1–B3) S, and (C1–C3) P before calcination and (A4), (B4), and (C4) after calcination at 350 °C. (**b**) TEM, HRTEM, and fast-Fourier-transform (FFT) images of the $Co_3O_4$ samples: (A1–A4) H, (B1–B4) S, and (C1–C4) P. Reprinted with permission from Ref. [77]. Copyright 2020 Elsevier.

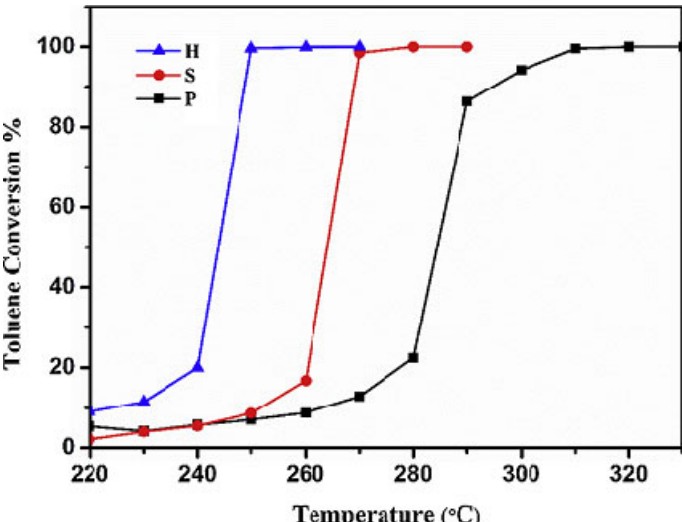

**Figure 8.** Catalytic performance of the synthesized $Co_3O_4$ as a function of reaction temperature for toluene oxidation under the following conditions: toluene 500 ppm, $O_2/N_2$ 20 vol%, total flow rate = 100 mL/min, WHSV = 60,000 mL/(g h). Reprinted with permission from Ref. [77]. Copyright 2020 Elsevier.

In another study [78], a $Co_3O_4$ mesoporous catalyst was synthesized and shown to be very active in removing traces of ethylene. This catalyst presented 30% ethylene conversion at 0 °C, compared with a nanosheet-type $Co_3O_4$ sample which did not present catalytic activity, even at 20 °C. The authors proposed that the outstanding catalytic activity of the first sample was due to the combination of the higher reactivity of the exposed facets (the {110} planes being more active than the {112} planes, mainly exposed by $Co_3O_4$ nanosheets) and its porous structure, which permitted the reactants to pass and be adsorbed into the pores, where they were chemically activated.

However, there have also been some studies that have found another sequence for the reactivity of the different exposed facets. For example, Yao et al. [69] obtained $Co_3O_4$ nanocatalysts with different morphologies and different exposed facets through hydrothermal routes: nanocubes with exposed {100} facets, hexagonal nanoplates with {111} facets, nanorods with {110} facets, and nanoplates with dominantly {112} exposed facets (Figure 9). In this research, the catalytic activities of the different facets were found to vary in the order {111} > {100} > {110} > {112}, and the superior activity of the {111} facet was attributed to the easier adsorption of propane. Moreover, the lattice oxygens in the $Co_3O_4$ nanoplates were more active than those of the other samples (as confirmed by $H_2$-TPR and $C_3H_8$-TPSR), and the activation of gas-phase oxygen on the $Co_3O_4$ hexagonal nanoplates surface was more facile, which is also beneficial for catalytic activity. However, the differences in reactivity, expressed as $T_{90}$ values, between the {111}, {100}, and {110} facets were, in this case, very small.

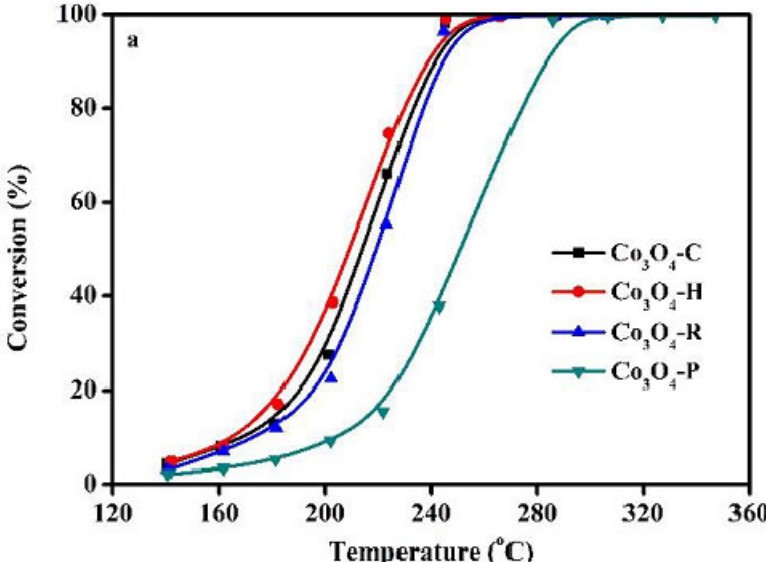

**Figure 9.** $C_3H_8$ combustion activities vs. temperature profiles of investigated catalysts. Reprinted and adapted with permission from Ref. [69]. Copyright 2019 John Wiley and Sons.

In other studies, it was found that the reactivity of different $Co_3O_4$ planes in propane oxidation: sometimes it was higher for {111} facets [79] and other times, for the preferable exposed {112} planes [80].

The contradictory results obtained in the literature are an indication that the problem of studying the factors that influence reactivity remains and that catalytic activity is determined by a combination of different parameters.

In Table 1, some of the results reported in the literature for the oxidation of propane and toluene, respectively, over cobalt oxides are summarized [35,36,55,59,61,63,65,69,70, 72,77,79,81–84]. The results include the specific reaction rate, measured at 200 °C and the temperature at which the conversion reaches a value of 90% (for measuring catalytic performance), the conditions of the reaction, and the main factors which were expected to determine the catalytic behavior of the oxidic materials: the specific surface area (BET), $Co^{3+}/Co^{2+}$ ratio, $O_{ads}/O_{latt}$ (both determined by XPS measurement), and the reduction temperature (determined by TPR measurements), as indicators of sample reducibility.

**Table 1.** Propane and toluene total oxidation over $Co_3O_4$.

| $CoO_x$ Sample | Reaction Conditions | BET $(m^2/g)$ | $Co^{3+}/Co^{2+}$ Ratio /$O_{ads}/O_{latt}$ Ratio | TPR Data $(T_{max})$ | $T_{90}$ (°C) | Reaction Rate $(\mu mol \cdot m^{-2} \cdot s^{-1})$ | Ref. |
|---|---|---|---|---|---|---|---|
| **Propane** | | | | | | | |
| $Co_3O_4$ nanoparticles | 8000 ppm $C_3H_8$ in air 50 mL/min | 47 | 0.47/0.52 | 352 | 235 | $0.14 \times 10^{-5}$ | [55] |
| $Co_3O_4$ nanoparticles | 1000 ppm $C_3H_8$ in air 100 mL/min | 48 | 1.28/0.52 | 107; 270; 366 | 260 | 0.017 | [81] |
| $Co_3O_4$ rods {110} | 2500 ppm $C_3H_8$ in air 30,000 mL/($g_{cat}$·h) | 48.5 | 0.46/0.57 | 113; 316 | 195 | $0.57 \times 10^{-5}$ | [70] |
| $Co_3O_4$ nanoparticles | $C_3H_8$:$O_2$:$N_2$ 1:10:89 100 mL/min | 120 | /0.78 | ~300; 350 | 240 [a] | 0.03 | [59] |
| $CoO$-$Co_3O_4$ | 0.3% $C_3H_8$ in air 30.000 mL/($g_{cat}$·h) | 62.8 | 0.65/1.47 | 333; 380 | 235 | - | [65] |

**Table 1.** *Cont.*

| CoO$_x$ Sample | Reaction Conditions | BET (m$^2$/g) | Co$^{3+}$/Co$^{2+}$ Ratio /O$_{ads}$/O$_{latt}$ Ratio | TPR Data (T$_{max}$) | T$_{90}$ (°C) | Reaction Rate (μmol·m$^{-2}$·s$^{-1}$) | Ref. |
|---|---|---|---|---|---|---|---|
| Co$_3$O$_4$ | 0.4% C$_3$H$_8$:20% O$_2$ in Ar 98 mL/min | 114 | - | - | ~210 | 0.015 | [36] |
| Co$_3$O$_4$ nanoparticles | 8000 vppm C$_3$H$_8$ in air 50 mL/min | 99 | - | 268; 313; 364 | ~250 [a] | - | [61] |
| Co$_3$O$_4$ nanoparticles | 1000 ppm C$_3$H$_8$ in air 100 mL/min 120.000 mL/(g$_{cat}$·h) | 82 | 1.16/1.00 | 104; 280; 370 | 225 | 0.01 | [82] |
| Co$_3$O$_4$ nanoparticles | 1000 ppm C$_3$H$_8$ in 21% O$_2$/N$_2$ 100 mL/min 40.000 mL/(g$_{cat}$·h) | 51 | - | 270 [b] | 224 | 32.9 × 10$^{-5}$ | [83] |
| Nanocubes {100} | 1% C$_7$H$_8$; 10% O$_2$/N$_2$ 33.3 mL/min 10.000 mL/(g$_{cat}$·h) | 50.2 | 1.20/0.78 | 317 | 242 | | [69] |
| Nanosheets {111} | | 41.6 | 1.14/0.85 | 303 | 239 | | |
| Nanorods {110} | | 43.9 | 1.39/0.69 | 310 | 245 | | |
| Nanoplates {112} | | 23.6 | 1.25/044 | 344 | 283 | | |
| Toluene | | | | | | | |
| Mesoporous (Kit6) | 1000 ppm C$_7$H$_8$ in O$_2$ (1/20 molar ratio) in N$_2$ 20.000 mL/(g$_{cat}$·h) | 121 | 1.46/0.95 | 283; 363 | 180 | - | [63] |
| Mesoporous (SBA16) | | 118 | 1.54/0.98 | 299; 382 | 188 | | |
| Mesoporous (SBA16) | 1000 ppm C$_7$H$_8$ in O$_2$ (1/200 molar ratio) in N$_2$ 33 mL/min | 313 | - | 273; 348 | 240 [a] | - | [35] |
| Co$_3$O$_4$ nanoparticles | 1000 ppm C$_7$H$_8$ in 21% O$_2$/N$_2$ 100 mL/min | 51 | - | 270 [b] | 242 | 150.8 × 10$^{-5}$ | [83] |
| ZIF-67, dodecahedral | 1000 ppm C$_7$H$_8$ in air 66.7 mL/min 20.000 mL/(g$_{cat}$·h) | 31.4 | 0.36/0.65 | 328; 395 | 254 | - | [84] |
| MOF-74, rods | | 32.2 | 0.45/0.83 | 300; 397 | 248 | | |
| ZSA-1, octahedral | | 63.4 | 0.57/1.17 | 294; 380 | 239 | | |
| Hydrangea {110} | 500 ppm C$_7$H$_8$, 20% O$_2$/N$_2$ 100 mL/min 60.000 mL/(g$_{cat}$·h) | 38.7 | 0.55/0.78 | 324; 396 | 248 | - | [77] |
| Spiky {100} | | 29.4 | 0.53/0.75 | 349; 455 | 269 | | |
| Pompon {111} | | 96.6 | 0.41/0.59 | 393; 482 | 298 | | |
| Flower {110} | 1000 ppm C$_7$H$_8$ in 20% O$_2$/N$_2$ 80 mL/min 20.000 mL/(g$_{cat}$·h) | - | 1.20/1.13 | 274; 344 | 228 | - | [72] |
| Rods {112} | | - | 1.02/0.69 | 287; 362 | 249 | | |
| 3D flower {111} | 1000 ppm C$_7$H$_8$, 20% O$_2$/N$_2$ 48.000 mL/(g$_{cat}$·h) | 84.6 | 1.73/0.94 | 227; 301 | 238 | - | [79] |
| 2D plate {112} | | 24.9 | 1.34/0.62 | 294; 385 | 249 | | |
| 1D needle {110} | | 24.9 | 1.26/0.51 | 308; 407 | 257 | | |

[a] T$_{100}$. [b] Lowest temperature peak.

## 4. Manganese Oxides as Catalysts in the Combustion Reaction

Manganese is one of the most abundant metals on the earth's surface. Its compounds are not expensive and are non-toxic, so they can be considered "environmentally friendly" materials.

Manganese is a transitional metal with five unpaired electrons in the d level (electronic configuration: [Ar] 3d$^5$4s$^2$), which allows it to adopt multiple oxidation states, hence its special properties. [85]. Manganese has the ability to form numerous stable stoichiometric oxides, MnO, Mn$_2$O$_3$, Mn$_3$O$_4$, and MnO$_2$ (Figure 10), but also metastable (Mn$_5$O$_8$) or unstable ones (Mn$_2$O$_7$—a green explosive oil).

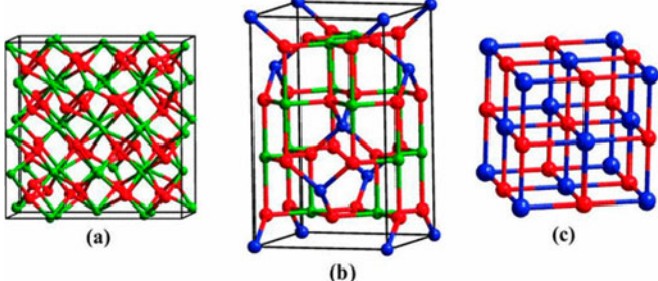

**Figure 10.** Crystal structures of mineral phases of (**a**) $Mn_2O_3$, (**b**) $Mn_3O_4$, and (**c**) MnO. All three manganese oxides are closely packed by lattice oxygen and differ only in the arrangement of octahedral and tetrahedral sites with various oxidation states of manganese. The green, blue, and red atoms represent $Mn^{III}$, $Mn^{II}$, and $O^{2-}$, respectively. Reprinted and adapted with permission from Ref. [86]. Copyright 2014 John Wiley and Sons.

The trivalent oxide, $Mn_2O_3$, occurs in two structural forms: $\alpha$-$Mn_2O_3$ (bixbyte mineral), stabilized as the body-centered cubic crystal structure; and $\gamma$-$Mn_2O_3$ (with spinel structure), which is less stable and not present in nature.

$Mn_3O_4$ (hausmannite) is a stable phase of manganese oxide and occurs as a black mineral. Hausmannite possesses a normal spinel structure, $Mn^{2+}Mn^{3+}{}_2O_4$, in which $Mn^{2+}$ and $Mn^{3+}$ ions occupy the tetrahedral and octahedral sites, respectively.

Manganese dioxide ($MnO_2$) can be found in the Earth's crust in massive deposits and in important deep-sea minerals. $MnO_2$ polymorphic crystallographic forms ($\alpha$-, $\beta$-, $\gamma$-, and $\delta$-) occur as a consequence of divergence in the connectivity of corner- and/or edge-shared octahedral [$MnO_6$] building blocks. $\alpha$-, $\beta$-, and $\gamma$-$MnO_2$ (also known as hollandite, pyrolusite, and nsutite, respectively) have tunnel structures, while $\delta$-$MnO_2$ (birnessite) has a layered structure composed of two-dimensional sheets of edge-shared [$MnO_6$] octahedra (Figure 11) [85,87].

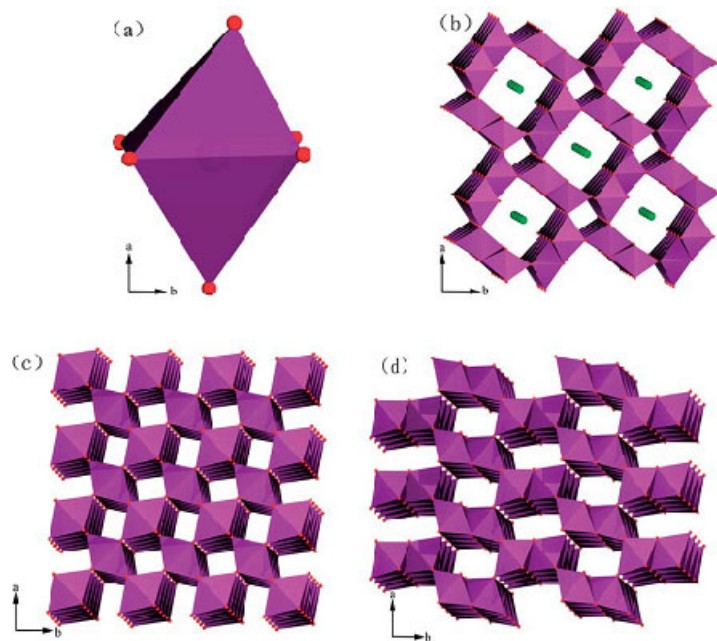

**Figure 11.** Polyhedral representations of the crystal structures of varieties of $MnO_2$: (**a**) [$MnO_6$] octahedron, (**b**) cryptomelane-type $\alpha$-$MnO_2$, (**c**) pyrolusite-type $\beta$-$MnO_2$, and (**d**) nsutite-type $\gamma$-$MnO_2$. Reprinted and adapted with permission from Ref. [87]. Copyright 2014 Royal Society of Chemistry.

$MnO_x$ with different oxidation states can either coexist or gradually become interconvertible from one form to another during the oxidation process controlled by the diffusion of oxygen. The strong ability to switch oxidation states and the formation of structural defects are beneficial to high oxygen mobility and oxygen storage, and are responsible for the catalytic properties of $MnO_x$ [31,88]. Due to their low cost, low toxicity, simple ease of preparation, and high stability, manganese oxides can be promising candidates for use in depollution catalysis. Among earlier publications about the catalytic properties of manganese oxides in hydrocarbon oxidation, the studies of Busca, Finocchio, Baldi, and their group [32,89–92] and the studies of Lahousse et al. [39,93] deserve special mention. They followed the total oxidation of propane over manganese oxides in various oxidation states, both pure and mixed oxides (with $TiO_2$, $Al_2O_3$, or $Fe_2O_3$) [89–92], and found that single Mn-based oxides were very active materials in the total oxidation of propane, propene, and hexane—generally more active than mixed oxides.

Existing types of manganese oxides, such as $MnO$, $MnO_2$, $Mn_2O_3$, and $Mn_3O_4$, have different structural characteristics which have a direct influence on their use as active catalysts for the oxidation of hydrocarbons. Numerous studies have been conducted in order to determine which of these oxides has the best performance in the oxidation of hydrocarbons. In a review of gaseous heterogeneous reactions over Mn-based oxides [31], it was claimed that $MnO_2$ usually has a higher catalytic oxidation performance than $Mn_2O_3$ or $MnO$, as was expected based on the MvK mechanism. However, there are other studies that have proved that catalytic activity does not always increase with the increasing oxidation state of manganese; different morphologies and different reaction conditions lead to different results. Therefore, Kim et al. [94] found that the activity of manganese oxides in the catalytic combustion of benzene and toluene on pure Mn oxides varied in the order $Mn_3O_4 > Mn_2O_3 > MnO_2$ and was related to the higher oxygen mobility and larger surface area of $Mn_3O_4$. Almost the same results were reported by Piumetti et al. [95], the sequence for the catalytic activity in the total oxidation of VOCs (ethylene, propylene, toluene, and their mixture) being: $Mn_3O_4 > Mn_2O_3 > Mn_xO_y$ (where $Mn_xO_y$ was a mixture of $Mn_3O_4$ and $MnO_2$). In this case, the higher activity of $Mn_3O_4$ was explained on the basis of the large number of electrophilic oxygen species adsorbed on its surface, proving their beneficial role in VOC total oxidation.

The superior catalytic performance of a metal oxide is closely related to its redox properties and hence to its nonstoichiometric, defective structure. The existence of $Mn^{3+}$ in a $MnO_2$ lattice usually generates oxygen vacancies and crystal defects, which increase its catalytic performance: the higher the oxygen vacancy density, the easier the activation adsorption of $O_2$. The formation of oxygen vacancies implies, also, the appearance of reduced Mn ions. The presence of both $Mn^{3+}$ and $Mn^{4+}$ has a beneficial effect because of the catalytic activity of $MnO_x$ generated by the electron transfer cycle between them (Equations (4) and (5)) [55,96]. Thus, in reference [96], various $MnO_x$ materials were synthesized using a coprecipitation method and then tested in the oxidation of toluene; in this case, the order of the catalytic performances of the obtained samples was also found to be directly related to the $O_{ads}/O_{latt}$ ratio values, which varied in relation to $Mn^{3+}/Mn^{4+}$ ratios.

The ability of manganese oxides to crystallize into various forms could be exploited to obtain catalysts with improved performances. There are many studies in the literature that have attempted to establish which is the most active crystallographic form.

Reference [93] reported on n-hexane oxidation over three crystallographic forms of manganese dioxides, namely, pyrolusite (β), nsutite (γ), and ramsdellite. The nsutite $MnO_2$ was proved to be about five times more active than the other forms in the reaction of VOC removal in terms of conversion. This behavior was explained by the increase in the oxygen lability in this case as a result of the appearance of Mn vacancies. The complete oxidation of n-hexane is a reaction demanding many oxygen atoms. So, each Mn vacancy (if it is situated close to the surface) offers six O atoms that could be easily removed to take an active part in the catalytic process. In another study [97], two manganese oxides, γ- and β-$MnO_2$, were synthesized and the γ-$MnO_2$ structure was also proved to be more active;

its activity was related to the existence of the $Mn^{3+}/Mn^{4+}$ couple and to the presence of lattice defects.

Huang et al. [98] synthesized $\alpha$-, $\beta$-, and special biphase (denoted as $\alpha@\beta$-) $MnO_2$ catalysts for toluene combustion (Figure 12a). $\alpha@\beta$-$MnO_2$ presented a larger specific surface area and higher oxygen vacancy concentration, which made it a better catalyst. The best performances were obtained for $\alpha$-$MnO_2$:$\beta$-$MnO_2$, with a mole ratio of 1:1 (Figure 12b).

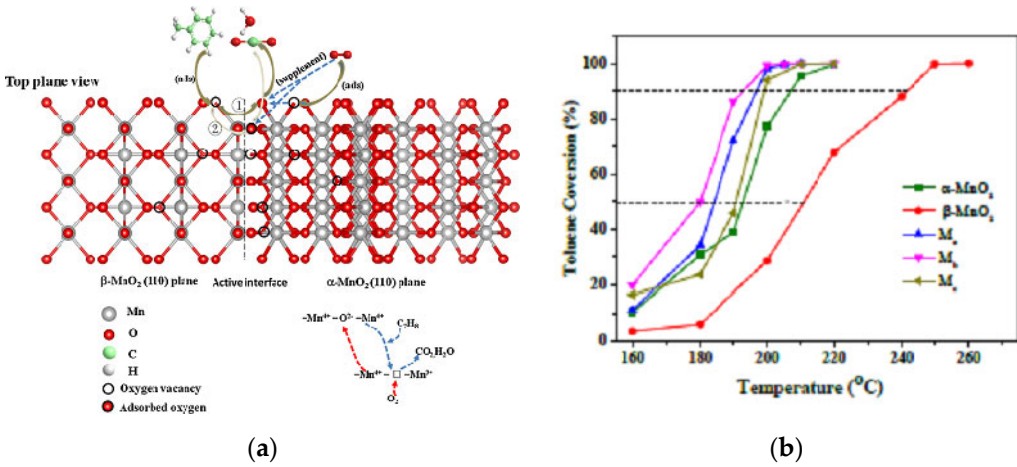

(**a**)          (**b**)

**Figure 12.** (**a**) Reaction scheme of toluene oxidation over $\alpha@\beta$-$MnO_2$ catalysts. (**b**) Activity profiles of the five samples for toluene oxidation as a function of temperature. Reprinted and adapted with permission from Ref. [98]. Copyright 2018 Elsevier.

In reference [99], the authors presented a schematic diagram for toluene oxidation over some $\alpha$-$MnO_2$ samples in order to highlight the role of the surface oxygen vacancies. The proposed mechanism was again MvK-type (Figure 13). The best catalytic performances were obtained over the catalyst with more surface oxygen vacancies and consequently more adsorbed oxygen, which supplemented the lattice oxygen consumed by the oxidation of toluene. On the other hand, more oxygen vacancies on the surface produced more $Mn^{3+}$ ions, which elongated the bond length of Mn-O and contributed to the easier release of surface lattice oxygen, which finally also led to better catalytic activity.

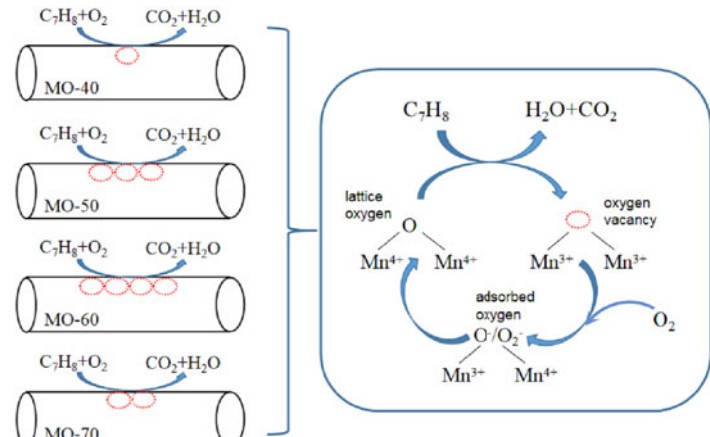

**Figure 13.** Mechanism of toluene oxidation on $\alpha$-$MnO_2$. Reprinted with permission from Ref. [99]. Copyright 2021 Elsevier.

Xie et al. [100], by a hydrothermal method, synthesized $MnO_2$ nanoparticles with $\alpha$-, $\beta$-, $\gamma$-, and $\delta$-crystal phases. $\alpha$-, $\beta$-, and $\gamma$-$MnO_2$ samples exhibited a 1D structure, and

δ-$MnO_2$ was a 2D layered-structure material. The catalytic activities of the $MnO_2$ samples decreased in the order of α- ≈ γ- > β- > δ-$MnO_2$. In this case, surface area was not the determinant in relation to catalytic activity, while δ-$MnO_2$ presented a higher surface area than the others; furthermore, a direct relationship between the reducibility and the activity of the studied samples could not be established, the easily reducible sample (δ-$MnO_2$ again) having the worst catalytic performance. The authors concluded that, in this case, the main influencing factors that determined the catalytic activity were morphology and crystal structure.

Controlling morphology is of great importance in order to obtain materials with desired properties. For this purpose, Wang et al. [101] prepared nanosized rod-, wire-, and tube- like morphologies for α-$MnO_2$ and flower-like morphologies for α-$Mn_2O_3$ and compared their physicochemical properties and catalytic activities in toluene combustion (Figures 14 and 15). The rod-like α-$MnO_2$ possessed the best catalytic activity for toluene combustion, mainly due to the higher amount of adsorbed oxygen and to the increased low-temperature reducibility. Their results were presented in Table 2.

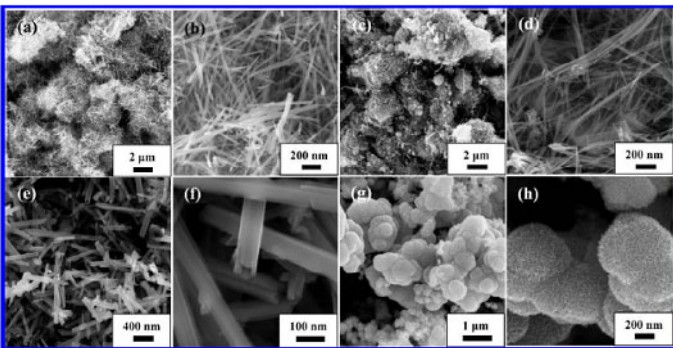

**Figure 14.** SEM images of (**a**,**b**) rod-like $MnO_2$, (**c**,**d**) wire-like $MnO_2$, (**e**,**f**) tube-like $MnO_2$, and (**g**,**h**) flower-like $Mn_2O_3$. Reprinted with permission from Ref. [101]. Copyright 2012 American Chemical Society.

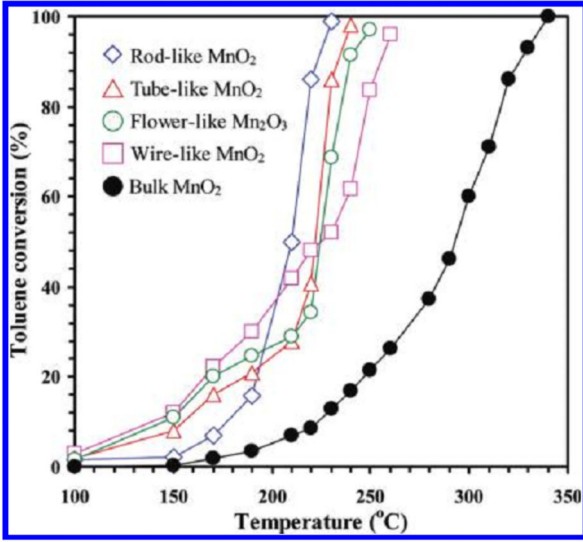

**Figure 15.** Toluene conversion as a function of reaction temperature over the catalysts under the conditions of toluene concentration = 1000 ppm, toluene/$O_2$ molar ratio = 1/400, and SV = 20,000 mL/(gh). Reprinted with permission from Ref. [101]. Copyright 2012 American Chemical Society.

In reference [102], the synthesis of hollow and solid polyhedral $MnO_x$ was reported. The hollow-structure materials proved to have outstanding performances in low-temperature

toluene combustion (100% conversion at temperatures up to 240 °C) and high thermal stability (Figures 16 and 17). Catalytic activity was correlated with cavity nature, the higher content of active oxygen, and the higher average oxidation state of Mn. From the TPR measurements, the solid sample presented two main reduction peaks centered at lower temperatures than for the hollow one, but the $H_2$ consumption in the second case was far higher (Figure 16). This is an indication that the hollow sample had much more active oxygen and might have been more reducible and active than the solid one (the consumption of hydrogen was directly related to the concentration of active oxygen).

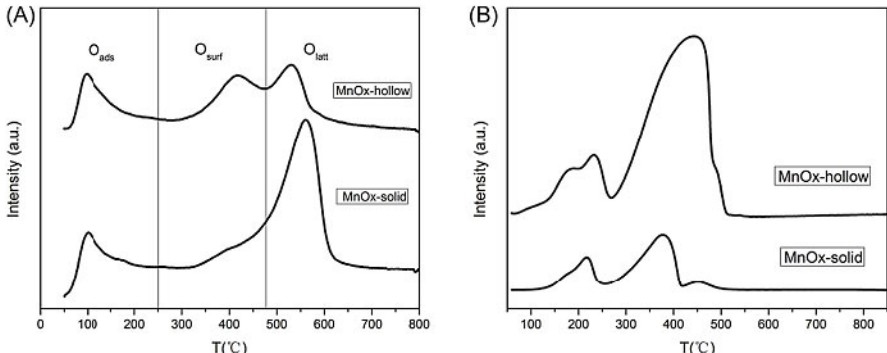

**Figure 16.** (**A**) $O_2$-TPD and (**B**) $H_2$-TPR profiles of MnOx-hollow and MnOx-solid. Reprinted with permission from Ref. [102]. Copyright 2017 Elsevier.

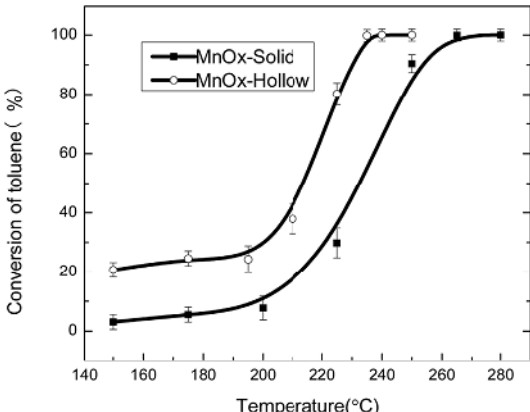

**Figure 17.** Toluene conversion over $MnO_x$ catalysts. Reaction conditions: catalysts weight = 0.3 g, toluene concentration = 1000 ppm, WHSV = 32,000 mL $g^{-1}h^{-1}$. Reprinted with permission from Ref. [102]. Copyright 2017 Elsevier.

A parameter of paramount importance for catalytic activity is the surface area of a catalyst, since the number of active sites increases as the exposed area increases. Thus, many authors have tried and succeeded in developing different methods for the synthesis of manganese oxides with increased surface areas [40,55,103]. In reference [55], $MnO_x$ with a high surface area and excellent activity in propane combustion was obtained ($T_{90}$ = 265 °C) using a wet combustion procedure in the presence of organic acids, which produced a significant increase in surface area. The best results were obtained with glyoxilic acid, the surface area increasing considerably from 3 $m^2$/g (in absence of organic acids) up to 43 $m^2$/g.

Reference [40] describes the successful preparation of mesoporous $\alpha$-$MnO_2$ microspheres with high specific surface areas, with control of the microstructures by adjustment of the synthesis temperature (Figure 18). When the temperature rose, the microspheres went from dense-assembly to loose-assembly, and the nanorod length on the surface of the $\alpha$-$MnO_2$ microspheres increased gradually. Among the catalysts prepared at different

temperatures, the best performance was exhibited by the catalyst with the larger surface area (Figure 19). A large surface area implies a higher concentration of surface $Mn^{3+}$ and consequently a higher concentration of oxygen vacancies and adsorbed surface oxygen species, which could be incorporated in the lattice or used themselves as active oxygen.

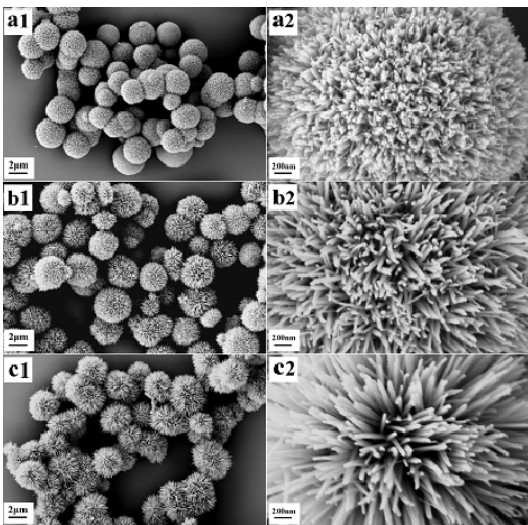

**Figure 18.** SEM images of $MnO_2$- 40 (**a**1,**a**2), $MnO_2$-60 (**b**1,**b**2), and $MnO_2$-80 (**c**1,**c**2). Reprinted with permission from Ref. [40]. Copyright 2016 Elsevier.

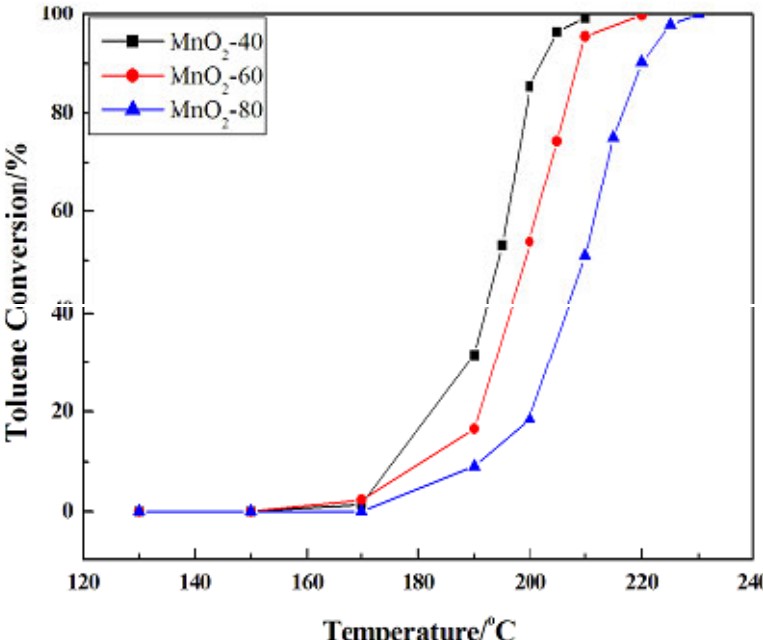

**Figure 19.** Catalytic combustion of toluene activity over $\alpha$-$MnO_2$ obtained at different temperatures. Reprinted with permission from Ref. [40]. Copyright 2016 Elsevier.

In another study [103], a new method was developed to obtain a material with a three-dimensional macroporous and mesoporous morphology and a $\gamma$-$MnO_2$-like structure by selectively removing La cations from three-dimensional ordered macroporous $LaMnO_3$. The as-prepared material had a high specific surface area (245.7 $m^2/g$) and excellent catalytic performance ($T_{90}$ = 252 °C, at GHSV = 120,000 mL $(g\ h)^{-1}$), which was related to both the three-dimensional macroporous and mesoporous morphology, which increased the

accessibility of toluene to the active sites, and the $\gamma$-MnO$_2$-like structure, which improved the O$_{latt}$ mobility of the material.

In Table 2, some results obtained for the oxidation of hydrocarbons over single manganese oxides are summarized, the temperature when 90% conversion (T$_{90}$) was reached being the measurement parameter for catalytic performance [40,88,90,93,95,96,99–102,104–108]. In several cases, the average oxygen state (AOS) of Mn ions instead of the Mn$^{4+}$/Mn$^{3+}$ ratio was listed. The value of AOS offers direct information about the reduction level of the oxide: the lower the AOS compared to the stoichiometric value, the more metallic ions in a reduced oxidation state exist in the oxide.

**Table 2.** Oxidation of hydrocarbons over single manganese oxides.

| MnO$_x$ | Oxidized VOC | Reaction Conditions | BET Surface (m$^2$/g) | Mn$^{4+}$/Mn$^{3+}$/O$_{ads}$/O$_{latt}$ | TPR | T$_{90}$ (°C) | Ref. |
|---|---|---|---|---|---|---|---|
| Mn$_2$O$_3$ | Naphtalene | 100 ppm VOC in air 50 mL/min GHSV: 60.000 h$^{-1}$ | 76 | - | 247,392 | 240 | [104] |
| Mn$_2$O$_3$ | Propane | VOC:air 0.5:99.5 50 mL/min GHSV: 15.000 h$^{-1}$ | 76 | - | 247,392 | 335 [a] | [105] |
| Mn$_2$O$_3$ | Toluene | 500 ppm VOC, 20% O$_2$ in N$_2$ 100 mL/min | 47.1 | 0.35/0.90 | 208,273,448 | 254 | [96] |
| Mn$_2$O$_3$ | Ethylene | 1000 ppm VOC in air GHSV: 19.100 h$^{-1}$ | 43 | 0.63/0.32 | 365,489 | 330 [b] | [95] |
| Mn$_2$O$_3$ | Toluene | 1000 ppm VOC 30% O$_2$ in Ar 50 mL/min | 7.7 | 0.90/0.63 | 379,434 | 270 [b] | [106] |
| Mn$_3$O$_4$ | Ethylene | 1000 ppm VOC in air GHSV: 19.100 h$^{-1}$ | 46 | 0.56/0.37 | 320,667 | 260 [b] | [95] |
| Mn$_3$O$_4$ | Propane Propene | 2% VOC; 12% O$_2$ in He 350 mL/min | 24 | - - | - - | 300 [b] 270 [b] | [90] |
| Mn$_3$O$_4$ hollow Mn$_3$O$_4$ solid | Toluene | 1000 ppm VOC in air WHSV: 32.000 mL/(g$_{cat}$·h) | 90 21 | 3.66 [c]/4.47 3.10 [c]/2.30 | 230,444 218,377 | 240 [b] 280 [b] | [102] |
| Mn$_2$O$_3$+MnO$_2$ | Ethylene | 1000 ppm VOC in air GHSV: 19.100 h$^{-1}$ | 43 | 1.28/0.25 | 350,370,489 | 350 | [95] |
| $\alpha$–Mn$_2$O$_3$+$\beta$–MnO$_2$ | Toluene | 4000 mgC/m$^3$ GHSV: 16.000 h$^{-1}$ | 46 | 3.78 [c]/0.31 | 320,360,430, 480 | 280 | [107] |
| MnO$_2$ rod MnO$_2$ wire MnO$_2$ tube Mn$_2$O$_3$ flower Bulk MnO$_2$ | Toluene | 1000 ppm VOC VOC/O$_2$ 1/400 in N$_2$ WHSV: 20.000 mL/(g$_{cat}$·h) | 83.0 83.2 44.8 162.3 10 | 1.72/1.50 3.22/0.78 2.04/1.15 0.24/0.92 - | 265; 285 275 268 260; 332 - | 225 245 233 238 322 | [101] |
| $\alpha$–MnO$_2$ | Toluene | 4 g/m$^3$ VOC GHSV: 10.000 h$^{-1}$ | 150 | 0.8/0.84 | 232 | 202 | [40] |
| $\alpha$–MnO$_2$ | Toluene | 1000 ppm VOC in air 66.6 mL/min WHSV: 60.000 mL/(g$_{cat}$·h) | 166 | -/0.72 | 294,348 | 267 | [40] |
| $\gamma$–MnO$_2$ $\beta$–MnO$_2$ | N-hexane | 250 ppm in air 100 mL/min | 100 50 | | | 200 260 | [93] |
| $\gamma$–MnO$_2$ | N-hexane | 125 ppm VOC, 20% O$_2$ in N$_2$ WHSV: 72.000 h$^{-1}$ | 103 | 3.9 [c] | - | 170 | [88] |

**Table 2.** *Cont.*

| MnO$_x$ | Oxidized VOC | Reaction Conditions | BET Surface (m$^2$/g) | Mn$^{4+}$/Mn$^{3+}$/O$_{ads}$/O$_{latt}$ | TPR | T$_{90}$ (°C) | Ref. |
|---|---|---|---|---|---|---|---|
| δ–MnO$_2$ | Toluene | 1000 ppm VOC, 21% O$_2$ in N$_2$ 150 mL/min WHSV: 90.000 mL/(g$_{cat}$·h) | 231 | 1.12/1.192 | 362 | 200 | [108] |
| α –MnO$_2$ | | 2000 ppm VOC 20% O$_2$ in N$_2$ 100 mL/min GHSV: 120.000 mL/(g$_{cat}$·h) | 245.7 | 1.02/0.97 | 349,401,469 | 252 | |
| β–MnO$_2$ | Toluene | | 163.4 | 1.67/0.34 | 364,406,475 | 258 | [103] |
| γ–MnO$_2$ | | | 11.3 | 1.30/0.25 | 349.401.469 | 304 | |
| δ–MnO$_2$ | | | 75.9 | 1.78/0.14 | 305,435 | 272 | |
| α–MnO$_2$ | Toluene | 1000 ppm VOC in air 500 mL/min GHSV: 30.000 h$^{-1}$ | 137.33 | 0.58/0.40 | 238,311,365, 376 | 203 | [99] |
| α–MnO$_2$ | | | 33.2 | - | 300 | 290 | |
| β–MnO$_2$ | Propane | 250 ppm VOC in air SV: 72.000 h$^{-1}$ | 8.4 | - | 322 | 373 | [100] |
| γ–MnO$_2$ | | | 64.3 | - | 324 | 280 | |
| δ–MnO$_2$ | | | 141 | - | 278 | - | |

$^a$ T$_{80}$. $^b$ T$_{100}$. $^c$ Average oxidation state (AOS).

Although an important factor to consider in the development of a high-performance catalyst is the stability of the catalyst, in this review we have not explored this vast topic, choosing instead to emphasize the redox properties. The poor resistance to SO$_2$ and H$_2$O has remained a challenge for some MnO$_x$-based catalysts, especially unsupported MnO$_x$ catalysts [109]. Tang et al. [41], by a chemical leaching method, synthesized a Co$_3$O$_4$ material with superior catalytic stability for propane oxidation at a high WHSV: 240.000 mL/(g h). Compared with a 1%Pd/Al$_2$O$_3$ sample (when the conversion decreased by more than 10%), for the Co$_3$O$_4$, any deactivation occurred within 60 h of measurement. This catalyst also had a very good water/sulfur-resistance property. At 280 °C, the conversion of propane remained complete for a long time when either 5 ppm SO$_2$ or 3.0% water was introduced. However, the water vapor effect on the catalytic oxidation of VOCs should be taken into consideration because it may adsorb on the active sites of porous materials and inhibit the adsorption of VOCs, suppressing their further catalytic oxidation [110,111]. The addition of water beyond an optimum level results in a loss of activity due to the sintering of the catalyst. Zhang et al. studied the stability of a Co$_3$O$_4$ nanocatalyst in the oxidation of toluene, with and without moisture [112]. The toluene conversions remained constant, without any deactivation under dry conditions during the stability test, but suffered a slight decrease (being 10% lower) when 1.5 vol.% H$_2$O was introduced into the stream. The addition of a higher concentration (5 vol.%) of moisture produced a sharp decrease in conversion, from 100 to 70%, though this rapidly came back to its original level once the moisture was stopped. This indicates that there was competitive adsorption between water and VOC molecules on the surface active sites, causing the deactivation of the catalyst, but it exhibited good regeneration ability and stability. Similar results regarding the effect of the concentration of water were also obtained by Li et al. [113]. Contrary to water, carbon dioxide is not an inhibitor, but the formation of carbonate species might lead to the deactivation of the catalyst when the reaction is performed at low temperatures [114]. On the other hand, the origin of carbonaceous material (coke) was ascribed to adsorbed carbon monoxide on the catalyst. The deposition of coke on the outer surface of catalysts or inside pores is another problem, because it can poison the active center and cause the deactivation of catalysts. Working in excess-oxygen conditions could prevent this problem.

In this review, we have limited ourselves to the study of CoO$_x$ and MnO$_x$ single oxides, considering this a starting point for such a vast field of research. MnO$_x$ particles supported on activated carbon, carbon nanotubes, Ce−Zr solid solutions, Al$_2$O$_3$, etc., are

used as catalysts for heterogeneous catalytic reactions [31]. $Al_2O_3$ and zeolite supports are considered inactive and enlarge surface areas, dispersing active sites. $CeO_2$ and $Ce-Zr$ solid solutions are active supports because they often serve as adsorption sites for reactions. Non-noble metal catalysts include mixed-metal-oxide catalysts and perovskite catalysts. Perovskite oxides (with the formula $ABO_3$) containing transition metals, such as Mn and Co, at B sites have high oxidation abilities toward VOCs [111]. $LaMnO_3$ (manganites) and $LaCoO_3$ (cobaltites) are considered active catalyst materials. A category of mixed oxides with improved properties is $CoMnO_x$. Han et al. prepared a $CoMnO_x$ catalyst with a high amount of surface-chemically adsorbed oxygen [115]; Dong et al. synthesized nanoflower spinel $CoMn_2O_4$ with a large surface area, high mobility of oxygen species, and cationic vacancies [116]. Various metals were used for composites with Mn or Co, because of the synergistic effect between the two metallic components, which can improve the catalytic properties of the final materials.

## 5. Conclusions

Due to the rapid growth of industrialization and urbanization, volatile organic compounds (VOCs) have become the main pollutants in air. Their abatement is imperative due to their harmful effects on human health and to their precursor role in the formation of photochemical smog. In this context, catalytic oxidation has proved to be a useful technology, as the pollutants can be totally oxidized at moderate operating temperatures under 500 °C.

High-performance noble-metal-based catalysts for complete oxidation of VOCs at low temperatures have been developed. However, these catalyst types present high manufacturing costs and poor resistance to poisoning. Hence, there is a need to replace these materials with abundant ones which are cheaper, less susceptible to supply fluctuations, and more environmentally friendly. Thus, a better alternative, widely explored, is the use of transition metal oxides as catalysts. They have the advantages of much lower costs, higher thermal stabilities, and resistance to poisoning, which are ultimately reflected in a decrease in the total cost of the depollution technology.

The oxidation of VOCs is a complex process that involves the consideration of many different parameters: firstly, there is a large variety of polluting organic compounds which interact differently with metallic oxides, depending on the structural and electronic properties specific to each class; on the other hand, interactions also depend on the structural and physicochemical properties of the catalyst itself. In this review, we targeted the oxidation of hydrocarbons, and the focus was on the characteristic properties of catalysts which influence their performance in total-oxidation reactions.

Based on the redox mechanism, which is accepted as the one that describes how the oxidation reaction of hydrocarbons on transition metal oxides generally proceeds, the catalytic activity of these oxides strongly depends on the following factors: (i) their specific surface areas and morphologies; (ii) their redox properties; and (iii) their abilities to adsorb oxygen in the form of high electrophilic species, $O_2^-$, $O^-$, which are also very active species for total oxidation.

Generally, the higher specific surface area, the larger the number of exposed active sites. In addition, a porous structure permits the diffusion of the molecules of reactants inside the pores, which leads to the easier accessibility of the active centers.

Redox properties are dependent on: (i) the presence of metallic ions in higher oxidation states ($Co^{3+}$, $Mn^{3+}$, or $Mn^{4+}$) which make the oxide easier to be reduced; (ii) the presence of oxygen vacancies, which act as the active centers for oxygen activation (necessary for re-oxidation of oxides after interaction with hydrocarbons); (iii) the presence of metallic ions in reduced oxidation states, which result in more oxygen vacancies on the surface and also produce the weakening of the bond length of M-O and contribute to the easier release of surface lattice oxygen; and (iv) the high mobility of lattice oxygen.

Cobalt and manganese oxides are abundant materials which are non-toxic and have been proved to be very active in the total oxidation of hydrocarbons, being good candidates

for the replacement of catalysts based on noble metals. In the case of cobalt oxides, $Co_3O_4$ was proved to be the most active, and the improvement of its catalytic performance has been achieved through controlled synthesis so as to obtain specific morphologies which present large surfaces and porous structures and which predominantly expose facets with high reactivity and defective structures (which contain large numbers of oxygen vacancies).

In the case of manganese oxides, the multitude of oxides with different oxidation states, which in turn can present several crystallographic phases, establishing the factors that determine their catalytic activities is more difficult. However, the increased reducibility, the presence of adsorbed oxygen, and the $Mn^{3+}/Mn^{4+}$ couple generally have a beneficial effect.

Briefly, in this review, recent progresses made in the development of single-oxide catalysts ($CoO_x$ and $MnO_x$) for the total oxidation of hydrocarbons has been summarized in order to establish the parameters that influence catalytic performances. Starting from this, new and high-performance catalytic materials with improved characteristics can be developed, either by obtaining specific morphologies, by depositing them on supports or by obtaining mixed oxides. Therefore, this research could be useful for tailoring advanced and high-performance catalysts for the total oxidation of VOCs.

**Author Contributions:** Conceptualization, V.B., P.C. and C.H.; writing—original draft preparation, V.B., A.V., P.C. and C.H.; writing—review and editing, V.B., A.V. and P.C.; supervision, V.B., A.V. and C.H. All authors have read and agreed to the published version of the manuscript.

**Funding:** This research received no external funding.

**Conflicts of Interest:** The authors declare no conflict of interest.

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
