# Peer review of "Insights into the Redox and Structural Properties of CoOx and MnOx: Fundamental Factors Affecting the Catalytic Performance in the Oxidation Process of VOCs"

_catalysts, doi:10.3390/catal12101134_

Round 1
Reviewer 1 Report
The authors submitted a manuscript of a review about the oxidation of VOCs over cobalt and manganese oxides. Although there are many reviews on this subject, I believe that the authors' approach is very interesting. They present a review of the literature from the perspective of explaining the source of the activity of cobalt and manganese oxides in the oxidation reactions of VOCs, discussing the properties of these oxides in detail.
In my opinion, reference to previous reviews (before 2000) on this subject is lacking. A reader may get the wrong impression that this research topic has aroused the interest of researchers only after the 2000s. Of course, it is impossible and useless to provide links to all the papers on this subject, but you could consider quoting two or more earlier reviews. In this way a less experienced reader (e.g. a student) can easily notice that researchers have been interested in this topic for many years (even before the EU directives appeared). Please take the above remark more as a hint or a point for thought (I realize that the more recent findings are of greater importance, nevertheless I was very pleased with the link to the original work by Mars va Krevelen).
I propose to remove the title of the chapter 'Results and discussion'. It seems to be inappropriate for the review article.
Figure 3 - what is Co-com? I guess this information is not included in the figure description?
Author Response
Thank you for your constructive comments. Point to point, the responses are in the attached file.

Reviewer 2 Report
In this review, the basic influencing factors of two different single oxide catalysts (CoOx and MnOx) catalysts on the oxidation activity of VOCs were summarized. For these two single oxide catalysts, their specific surface area, oxidation state of metal ions, oxygen vacancies, crystal defects, chemisorbed oxygen/lattice oxygen mobility and catalyst morphology jointly affect the catalyst activity. Among all the above factors, the morphological part is slightly different, Co3O4 summarizes the different oxidative activities of VOCs with different exposures, and MnO2 focuses on summarizing different crystal forms with different catalytic activities for VOCs. This review has clear ideas, detailed content, and a lot of work. However, there are still some details that need to be dealt with. In addition, the logic of this review needs to be sorted out, and the questions need to be addressed before publishing on Catalysts.
1. Some details should be noted:
① “–M(n-1)+–â–¡–M(n-1)+– + 1/2O2 → –Mnn+–O2-–Mnn+ ”(formula (5), line 122)
② “The best catalyst, sample H, performed a 90% toluene conversion at about 248℃, which was 50 °C lower than that of sample P (Figure 7).” (line 288-289)
③ ** is not marked in Table 1.
④ “(c) pyrolusite-type β-MnO2 and (d) nsutite-type γ-MnO2”. (line 384)
⑤ Replace T100 in Table 2 (line 9) with **
⑥ Add the serial number of the depicted figure at the end of the article description paragraph (figure 11, 13-18). Eg: “The best performances were obtained for a mole ratio of 1:1 (α-MnO2:β-MnO2) (Figure 11). ” (line 445-446)
⑦ Figure 11(b): Replace the serial number (a) (marked in the picture) with (b).
⑧ “Figure 12. Mechanism of toluene oxidation on α-MnO2 . (from Ref. [88])” (line 461)
⑨ “Wang et al. [83] prepared nanosized rod-, wire-, tube-like morphologies α-MnO2, and flower-like morphologies α-Mn2O3 and compare their physicochemical properties and catalytic activities in toluene combustion (Figure 13, 14).” (line 474-476)
⑩ “Figure 18. Catalytic combustion of toluene activity over α-MnO2 obtained at different temperature. (from Ref. [37])” (line 529-530)
2. It is recommended to add specific surface area and specific morphology to the influencing factors of the abstract.
3. “Co-com” in the last column of the abscissa in Figure 3 appears for the first time, please explain the meaning of the abbreviation in the article; the characterization results of “Co-com” are similar to the “Co-CO3”, please explain why the catalytic activity of "Co-com" in Figure 3 is not outstanding only for toluene.
4. It is recommended that the figure notes in the quoted figures be more clearly labeled (Figure 5(b), Figure 11(b)).
5. “S sample even possessed the same primarily exposed {110} facets proved to have lower activity” (line 291), are the samples named “Spiky {100}” appearing in Table 1 inconsistently named?
6. The first appearance of “*** - average oxidation state (AOS)” (line 419) in the table notes of Table 2 suggests a brief explanation.
7. “a higher valence of Mn it is expected to be favorable.”(line 397) , “The existence of Mn3+ in the MnO2 lattice usually generates the oxygen vacancies and crystal defects, which raise its catalytic performances”(line 417-418), “The presence of both Mn3+ and Mn4+ has a beneficial effect because the catalytic activity of MnOx was generated by the electron transfer cycle between them”(line 421-423); it is recommended to add logical correlations between these three sentences.
8. It is recommended that the order of the article description follow the order of the conclusions.
Author Response

(The authors gave the same response as above.)

Reviewer 3 Report
The review is very complete and useful to obtain a broad knowledge of CoOX and/or MnOx catalysts for the combustion of VOCs. Different aspects related to the activity of these single oxide-based catalysts are addressed. The main factors affecting the catalytic performance of several metal oxide catalysts, CoOx and MnOx, respectively, towards total oxidation of hydrocarbons, have been reviewed. These factors include redox properties, morphology, surface defects, and lattice oxygen mobility, among others.
Although this review could be accepted in this form, the authors are suggested to add some comments regarding the following points:
1-Generally, the proposed mechanism of VOCs combustion includes the Eley-Rideal (E-R), Langmuir-Hinshelwood
(L-H) and Mars-van-Krevelen (MVK) models. Could the authors indicate some detail of the mechanisms not mentioned in the review?
2- Could the authors include information about the potential use of these oxides in supported and/or structured catalytic systems?
Author Response

(The authors gave the same response as above.)

Reviewer 4 Report
The authors briefly reviewed the main factors affecting the catalytic performance of several metal oxide (e.g., CoOx and MnOx) catalysts for the total oxidation of volatile organic compounds (VOCs), discussed the relationship of their redox properties, nonstoichiometric, defective structure, and with their lattice oxygen mobility with catalytic performance, and envisioned the future work on tailoring advanced and high-performance catalysts for the total oxidation of VOCs. This work contains some useful information and could be considered for publication. However, the authors should revise their manuscript before acceptance for publication according to the following comments:
1. In addition to the Mars van Krevelen mechanism, there are other mechanisms working in the oxidation of VOCs over transition metal oxides.
2. The surface adsorbed oxygen species play an important role in catalyzing the oxidation of VOCs. How do the authors consider the role of the surface adsorbed oxygen species?
3. The kinetics results are suggested to be also discussed in this review work.
4. In all of the tables, T90 is meaningless since VOCs concentrations, space velocities, catalyst mass are different over these catalysts cited in the tables. It would be better to use specific reaction rates at a certain temperature to compare catalytic activities of the catalysts.
5. How about the catalytic stability of the typical catalysts in the presence of water, carbon dioxide or even sulfur dioxide?
6. There are some inappropriate English words or expressions in the manuscript. The authors should carefully polish the English of the whole manuscript.
Author Response

(The authors gave the same response as above.)

Round 2
Reviewer 4 Report
I have carefully checked the responses and modifications of the revised manuscript, and found that the authors have properly modified their manuscript according to the Reviewers' comments. Hence, I think that it is now acceptable for publication.